

# The evolution of resource management in Taiwanese fisheries: coastal and offshore perspectives

Yan-Lun Wu[1,*], Irene Chia Ling Lim[1,*], LiXliang Li[1], Lu-Chi Chen[2], Po-Yuan Hsiao[1], Wei-Yu Lee[1] and Kuo-Wei Lan[1]

[1] Center of Excellence for Oceans, National Taiwan Ocean University, Keelung, Taiwan, R.O.C.
[2] Marine Fisheries Division, Fisheries Research Institute, Ministry of Agriculture, Keelung, Taiwan
* These authors contributed equally to this work.

## ABSTRACT

This study investigates the historical changes in resource development of offshore and coastal fisheries around Taiwan from 1970 to 2021 using the mean trophic level (MTL) and Fishing-in-Balance (FiB) indices. Utilizing data from the Fisheries Statistical Yearbook, three scenarios were employed to assess the effects of highly migratory, oceanic migratory, and seasonal migratory fish species on the MTL and FiB indices. The analysis revealed a continuous increase in MTL for offshore and coastal fisheries, suggesting shifts in the trophic structure and potential over-exploitation of higher trophic level species. The FiB index for offshore fisheries exhibited a declining pattern after 1990, reflecting the depletion of indigenous fish stocks, while the FiB for coastal fisheries showed an increasing trend from 1970 until the 2000s, followed by a decrease, highlighting unsustainable fishing practices. Moreover, the study identified *Scomber australasicus*, targeted by Taiwanese Purse Seine, as an influential species affecting the resource dynamics of offshore fisheries. The trawling fisheries would also be a crucial issue affecting the indigenous resource dynamics of offshore fisheries in Taiwan. This study identified a fishing-down mechanism within indigenous fishery dynamics. The potential over-exploitation of fish stocks could result in long-term unsustainable practices if left unaddressed. This study advocates for enhanced fisheries management through stricter regulations on fishing gear, continuous monitoring, and adaptive management strategies. These measures are essential for achieving sustainable development goals and conserving marine biodiversity in Taiwan's waters. By addressing these critical issues, Taiwan can better manage its fisheries resources and promote ecological balance.

# INTRODUCTION

## The goal for coastal and offshore fisheries around Taiwan linked to SDG

The Sustainable Development Goal (SDG) indicators are tools that help monitor progress, enabling governments, organizations and stakeholders to track the advancement towards

Corresponding author
Kuo-Wei Lan,
kwlan@mail.ntou.edu.tw

the 17 Sustainable Development Goals set by the United Nations (*Virto, 2018*; *Nash et al., 2020*; *Moffitt & Cajas-Cano, 2014*). SDG 14- Life Below Water, highlights the necessity to conserve and sustainably use oceans, seas, and marine resources for sustainable development (*Virto, 2018*). One of the crucial targets is protecting and restoring ecosystems (14.2) (*Nash et al., 2020*) which is linked to "Number of countries using ecosystem-based approaches to managing marine areas" (14.2.1) (*Virto, 2018*). The indicator aims to promote sustainable marine management of marine environments in order to mitigate adverse impacts (*Virto, 2018*; *Moffitt & Cajas-Cano, 2014*).

Sustainable development has also been a priority in Taiwan. Taiwan has developed its own SDGs taking into reference the United Nations SDGs from 2016 (*Ho et al., 2022*). The fisheries resources of Taiwan have been influenced by several variables such as technological advancements, economic shifts, and environmental considerations (*Liao, Huang & Lu, 2019*). The Fisheries Agency of Taiwan has collected and summarized national fisheries statistics into yearbooks since 1959, and a separate set of statistics is also available (*Kuo & Booth, 2011*). The fisheries data encompass catches from three distinct fisheries types. First, the 'coastal' fisheries, which are artisanal fisheries operating up to 12 nautical miles off shore. Second, the 'offshore' fisheries, which are industrial fisheries operating within Taiwan's exclusive economic zone (EEZ). Third, the 'far seas' fisheries which are distant water industrial fisheries operating outside the EEZ of Taiwan.

## The current status and challenges of coastal and offshore fisheries around Taiwan

The complexity of fishing gear is driven by the diverse habitat characteristics found in Taiwan's coastal and offshore fisheries around Taiwan (*Liu, 2013*; *Liao, Huang & Lu, 2019*). The diverse fishing methods and gears utilized in fisheries significantly impact the species composition and distribution in different habitats and migratory pattern (*Barnette, 2001*; *Lin et al., 2023*; *Naimullah et al., 2023*). This is due to the selectivity of the fishing gear, which ranges from small-scale coastal fishing to large- scale offshore operations. For example: coastal fishing method such as purse seines were commonly used since 1980 to catch grey mullet fish, however, studies revealed the shift in fishing gear usage to gillnets and traps after 2000 (*Lan et al., 2017*). For offshore fisheries around Taiwan, longline was the most common gear used to target highly migratory species such as tuna and billfish while trawling was commonly used to targets migratory pelagic fish such as mackerel and squid, but also included a high proportion of bycatch of sharks, sea turtles and other non-targeted species (*Huang & Liu, 2010*). This highlights the importance of incorporating the effects of fishing gear selectivity and habitat characteristics when evaluating the dynamics of fishery resource.

Coastal and offshore fisheries around Taiwan face numerous challenges and difficulties threatened by several issues including overfishing, climate impacts and limited integration of scientific research (*Parsons et al., 2014*). Overfishing has significantly depleted key fish stocks, reduced the biodiversity and compromising the sustainability of fisheries (*Jagers, Berlin & Jentoft, 2012*; *Liao, Huang & Lu, 2019*). Inadequate enforcement of fishing regulations and ineffective management strategies contribute to resource depletion, while

limited integration of scientific research into policy-making hinders adaptive management practices (*Chen, 2012*; *Huang et al., 2016*; *Liao, Huang & Lu, 2019*). The lack of consensus on the definition of ecosystem-based management further complicates efforts to promote fishery sustainability in Taiwan (*Garcia & Cochrane, 2005*).

## The way to clarify the development of fishery resource

Previous studies have indicated that data collection and scientific assessment are necessary initial steps to achieve sustainable coastal and offshore fisheries around Taiwan (*Liao, Huang & Lu, 2019*). Analyzing the mean trophic level (MTL) can gain a deeper understanding of the ecological consequences of fishing activities and help identify potential areas for conservation and management intervention (*Pennino et al., 2011*). MTL is calculated using species biomass data from different sources and used to monitor the mean trophic level of fisheries catches within an ecosystem, providing insights into the overall food web structure (*Tolimieri et al., 2013*; *Kleisner, Mansour & Pauly, 2014*). It effectively monitors fluctuations in the mean trophic level of a variety of exploited species, offering a window into how fishing pressure impacts marine biodiversity.

However, declines in trophic level can be masked by geographic expansion and/or the development of offshore fisheries (*Kleisner & Pauly, 2011*). The Fishing-in-Balance (FiB) index was created to serve as a metric for assessing variations in the size of fishing fleets over time, as indicated by the trophic levels of the catches (*Pauly, Christensen & Walters, 2000*; *Bhathal & Pauly, 2008*; *Kleisner & Pauly, 2011*; *Kleisner, Mansour & Pauly, 2014*). The "Sea Around Us" project, a global research initiative, also employs these indices to assess the impact of fisheries on marine ecosystem worldwide (*Kleisner, Mansour & Pauly, 2014*). This project provides a comprehensive database and analytical tools that offer valuable benchmarks for understanding local fishery dynamics within a global context, aiding in the comparison and validation of regional studies (*Kleisner & Pauly, 2011*; *Kleisner, Mansour & Pauly, 2014*).

Previous studies have shown that the fishery resource carrying capacity of Changshan Islands around China exhibited overfishing status when evaluated by MTL indicators (*Cao, Sun & Yang, 2023*). The fishing in the Central Amazon showed unsustainable status and would benefit from better management strategies through the MTL and FiB index (*Matos et al., 2024*). *Su et al. (2021)* also suggested to reduce fishing pressure and strengthening the implementation of protection measures for South China Sea examined by MTL and FiB indicators. To summarize, investigating the local fishery resources carrying capacity and fishing statement by using MTL and FiB indicators is considered crucial and directly ways to achieve sustainable fisheries management.

## Objective

This study aims to collect the Fisheries Agency (FA) Fisheries Statistical Yearbook from 1970 to 2021 to investigate the fishery resource dynamic of coastal and offshore fishery around Taiwan. However, external activities (*e.g.*, fishing behavior, policy implementation) can cause a preference for capturing specific species (*e.g.*, tuna, sharks, or pelagic species), which in turn leads to external factors affecting regional fishery resource assessment

(*Fulton et al., 2011*). Previous research has also shown that analyzing the MTL at different threshold levels can help identify the underlying causes of trends in this indicator, particularly when examining pelagic and demersal species separately (*Arroyo et al., 2019*). The regional variations of MTL would be an issues worthy of attention although Sea Around Us have developed the regional MTI to address this situation (*Kleisner, Mansour & Pauly, 2014*). The MTL for each local region can then be calculated, with subsequent regions determined sequentially (*Kleisner, Mansour & Pauly, 2014*). In Taiwan, various external factors (*e.g.*, valuable species) and the characteristics of fish (*e.g.*, seasonal or oceanic migratory species) play a crucial role in influencing the fishery resource dynamics. To address these issues, three scenarios are proposed for evaluating the offshore and coastal fishery dynamics around Taiwan. Each scenario is designed to refine the understanding of how different species contribute to the overall fishery dynamics. The study employs key ecological indicators, such as the MTL and FiB indices to provide insight into the ecological impacts of fishing activities and the balance of fisheries over time. These indicators are calculated using decades of historical data from the Fisheries Statistical Yearbook, enabling a comprehensive historical analysis. Additionally, this study aims to identify the influential species affecting fishery dynamics to assess the current state of fisheries in comparison to the past several decades.

## MATERIALS AND METHODS

### Data collection

The Fisheries Statistical Yearbook from 1970 to 2021 were collected from the FA, Ministry of Agriculture of Taiwan. The initial dataset included information on fishing gear, fishing area, and catch (kg), categorized into coastal (within 12 nautical miles) (Fig. 1A) and offshore (12–200 nautical miles) fisheries (Fig. 1B; *Fisheries Agency, 2020*). The number of powered craft increased from 11,365 to 12,182 fleets from 1972 to 2021. However, the dataset on the number of powered craft categorized by fishing gear is only recorded by tonnage (*Fisheries Agency, 2021*). There is no data available on the number of powered craft categorized by coastal and offshore fisheries for each fishing gear. Therefore, the number of fleets is not included in this study.

The trophic level ($TL_i$) and habitat characteristics of each species were obtained from feeding studies and information from the FishBase (https://www.fishbase.se/search.php), which provides TL of each species by estimating using data from stomach content analysis and stable isotope studies, based on food source of the fish species (*Froese & Pauly, 2000*). Initially, the dataset was screened for consistency in species names and classification. A comprehensive analysis was ensured by including species consistently recognized despite changes in nomenclature and species name (*Costello et al., 2013*). The data was further filtered to include species with reliable trophic level information. Species that were difficult to differentiate or for which trophic level information was challenging to obtain, particularly closely related species, were excluded. For instance, small anchovy like the Shorthead anchovy (*Encrasicholina heteroloba*), Japanese anchovy (*Engraulis japonius*) and other larval fish with 700 tons catches in 2021, took around 3.5% for total catches (19,863 tons) for coastal fisheries, can be easily misidentified owing to insufficient
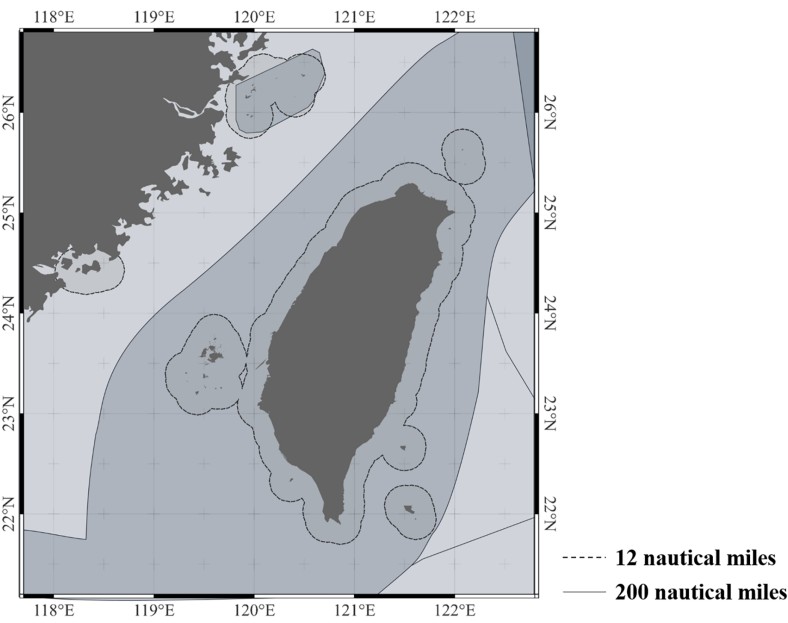

**Figure 1 The study area of this study: coastal fisheries of Taiwan with dashed line (within 12 nautical miles) and offshore fisheries of Taiwan (12–200 nautical miles) with solid line.** The maps are generated by QGis (https://www.qgis.org). The range of 200 nautical miles (EEZ) are plotted according to FAO (https://www.fao.org/home/en).

morphological diagnostic characters (*Ko et al., 2013*; *Afrand et al., 2020*; *Fisheries Agency, 2021*). Due to this potential for misidentification, anchovies were excluded to ensure the reliability and accuracy of the findings. In addition, marine organisms like squid, bivalve, lobsters, and crabs were not included in the study. This exclusion is due to biases reported in the trophic level assessment of squid (*Logan & Lutcavage, 2013*; *Ibáñez et al., 2021*) and insufficient information available for crabs and bivalves (*Oesterwind & Piatkowski, 2023*), which complicate accurate analysis (*Wheeler & Stebbing, 1978*; *Hussey et al., 2014*). Fish species, in contrast, have more consistent and well-documented data, presenting acrossalmost all trophic level and provide reliable indicators of the impacts on fishing (*Shannon et al., 2014*). Overall, the catches and trophic levels of the top 100 species were utilized (Table S1).

## MTL calculation

The MTL for the fisheries catch data of each year $j$ was computed using the initial formula introduced by *Pauly et al. (1998)*:

$$MTL_j = \sum_i \left( TL_i \times Y_{ij} \right) / \sum_i Y_{ij}$$

where $TL_i$ represents the trophic level of the species or taxonomic group $i$, and $Y_{ij}$ denotes the catch of the species or taxonomic group $i$ in the year $j$ (*Pauly & Palomares, 2005*). This approach provides information about the composition of the catch in terms of the trophic levels of the species or taxonomic groups that make up the overall catch for a given time period (*Pauly & Palomares, 2005*).

## FiB calculation

The trophic level of catch was examined in relation to geographic growth using the FiB index. The index was designed to remain constant even when a decrease in the mean trophic level is offset by an increase in catch volume. This aligns with the energy flows in ecosystems being pyramidal in character and the general 10% transfer efficiency between trophic levels (*Pauly, Christensen & Walters, 2000*). The FiB index was determined using the following formula:

$$FiB = \log\left( Y_i \left(\frac{1}{TE}\right)^{TL_i} \right) - \log\left( Y_0 \left(\frac{1}{TE}\right)^{TL_0} \right)$$

where $Y_i$ is the total catch at year $i$, $TL_i$ is the mean TL of the total catch at year $i$, TE is the trophic efficiency (set at 0.10), and $Y_0$ and $TL_0$ are the total catch and mean TL of the first year of the time series. The FIB index was used as an indicator of a "trophic level balance" in fishery management, assessing whether the fisheries are ecologically balanced. FiB value < 0 may indicate unbalanced fisheries, where the current catch is lower than the theoretical catch that would be expected based on trophic structure of the ecosystem (*Pauly, Christensen & Walters, 2000*; *Christensen et al., 2011*).

## The induction of each scenario

Although the MTL and FiB index for all species that are mentioned in Table S1 were calculated, the Taiwanese fishermen primarily target commercial significant migratory species. Consequently, this study evaluates the dynamics of water of Taiwan by several scenarios. The Scenario All encompasses all species recorded in the fisheries statistics, excluding some significant species which are mentioned in "Data collection" (Table S1). Subsequently, this study systematically refines the list of the species by excluding certain species or family in three scenarios. The data description of each scenario and the total catch of the fisheries of coastal and offshore fisheries data after exclusion process of each scenario were listed in Table 1.

First, tuna species (*e.g.*, Yellowfin tuna, Bigeye tuna) were excluded in Scenario 1 because these species are highly migratory species that possess significant economic value. These species are heavily targeted by commercial fisheries due to their demand in the global market. Scenario 2 involved the exclusion of oceanic migratory species (*e.g.*, Marlin, and Swordfish). These species exhibit pronounced seasonal variations in their migratory patterns, their presences in the water of Taiwan can fluctuate significantly throughout the year.

Scenario 3 involved the removal of seasonal migratory species (*e.g.*, Grey mullet and Mackerel). These species were not extensive migratory species as in Scenario 2, however, their migratory behavior coupled with seasonal spawning activities, introduces variability that can obscure trends and patterns in the local fish population. After going through all the scenario, the remaining species comprised species that indigenous in the water of Taiwan. The indigenous species are relatively stable and localized presence in the water of Taiwan. The detail information about the species excluded in each scenario is

**Table 1 Description of each scenario and the total catch of the fisheries of coastal and offshore fisheries data after exclusion process of each scenario.**

| Scenarios | Description | The total catch of coastal fisheries data (%) | The total catch of offshore fisheries data (%) |
|---|---|---|---|
| Scenario all | All species | 100% | 100% |
| Scenario 1 | Scenario all exclude tuna species | 90% | 89.8% |
| Scenario 2 | Scenario 1 exclude oceanic migratory species | 73.9% | 72.9% |
| Scenario 3 | Scenario 2 exclude species with seasonal migration | 23.1% | 22.6% |

**Table 2 The species removed from each scenario for offshore and coastal fisheries.**

| Scenarios | Description | Scientific name/family group |
|---|---|---|
| Scenario 1 | Scenario a exclude tuna species | *Katsuwonus pelamis, Thunnus alalunga, Thunnus albacares, Thunnus obesus,* and *Thunnus orientalis* |
| Scenario 2 | Scenario 1 exclude oceanic migratory species | Carcharhinidae, *Cololabis saira, Coryphaena hippurus, Cypselurus unicolor, Euthynnus affinis, Istiompax indica, Istiophorus platypterus, Isurus oxyrinchus, Kajikia audax Makaira mazara, Mola mola, Prionace glauca* and *Xiphias gladius* |
| Scenario 3 | Scenario 2 exclude species with seasonal migration | *Aluterus monoceros, Auxis* spp., *Carcharhinus limbatus, Cheilopogon unicolor, Decapterus maruadsi, Hemitriakis japonicas, Lutjanus argentimaculatus, Megalaspis cordyla,* Monacanthidae, *Mugil cephalus, Polydactylus sextarius, Rachycentron canadum, Ruvettus pretiosus, Sardinella sindensis, Scomber australasicus, Scomberomorus* spp., *Sphyraena barracuda, Trachurus japonicas, Trichiurus lepturus* |

comprehensively outlined Table 2. Following this process, the list of the indigenous species was compiled in Table 3.

**Contribute trophic level (CTL) calculation**

In addition, the MTL is influenced by the trophic levels of the various fish species caught. The changes in MTL caused by a single fish species would require calculating the percentage contribute to the trophic level (CTL) of that single species to the MTL. The calculation formula is as follows:

$$CTL_i = \frac{TL_i Y_{ij}}{\sum TL_i Y_{ij}} \times 100\%$$

where $CTL_i$ is the contribute trophic level of the species or taxonomic group $i$, and $Y_{ij}$ is the catch of the species or taxonomic group $i$ in the year $j$, and $TL_i$ is the trophic level of the species or taxonomic group, determined as describe previously. This calculation takes into account both the trophic level of the species and its catch in a specific year, allows us to comprehend how individual species affect variation in MTL over time.

This study only examines the CTL for the coastal and offshore fisheries data in Scenario 2 and Scenario 3, as presented in Table 1. The purpose is to determine the species-trophic-level contribution to the overall trophic dynamics. For Scenario 2, which excludes the oceanic migratory species, the CTL analysis is important to identify the key species that have a significant influence and contribution to the trophic structure. This is crucial, as the

**Table 3 Description of indigenous species around water of Taiwan that remain after exclusion of several scenarios.**

| Family name | Scientific name | Common name | Trophic level |
|---|---|---|---|
| Niphonidae | *Niphon spinosus* | Ara | 4.5 |
| Carangidae | *Seriola dumerili* | Greater amberjack | 4.5 |
| Synodontidae | *Saurida elongata* | Slender lizardfish | 4.5 |
| Chirocentridae | *Chirocentrus dorab* | Dorab wolf herring | 4.4 |
| Sphyraenidae | Sphyraena | Great barracuda | 4.4 |
| Muraenidae | *Enchelycore schismatorhynchus* | White-margined moray | 4.38 |
| Caranginae | *Carangoides malabaricus* | Malabar trevally | 4.36 |
| Psettodidae | *Psettodes erumei* | Indian halibut | 4.35 |
| Lutjanidae | *Lutjanus jocu* | Dog snapper | 4.35 |
| Rajidae | *Amblyraja hyperborea* | Arctic skate | 4.3 |
| Epinephelidae | *Epinephelus malabaricus* | Malabar grouper | 4.2 |
| Priacanthidae | *Priacanthus macracanthus* | Red big-eye | 4.11 |
| Lutjanidae | *Lutjanus bohar* | Two-spot red snapper | 4.1 |
| Sciaenidae | *Pennahia macrocephalus* | Big-head pennah croaker | 4.08 |
| Muraenidae | – | Moray eels | 4.07 |
| Sciaenidae | *Pennahia argentata* | Silver croaker | 4.06 |
| Synodontidae | Saurida | Lizardfishes; Bombay ducks | 4 |
| Epinephelidae | *Epinephelus coioides* | Orange-spotted grouper | 4 |
| Latidae | *Psammoperca waigiensis* | Sand bass | 4 |
| Dasyatidae | Bathytoshia lata | Brown stingray | 4 |
| Epinephelidae | *Epinephelus lanceolatus* | Giant grouper | 4 |
| Sciaenidae | *Nibea albiflora* | Yellow drum | 4 |
| Centrolophidae | *Psenopsis anomala* | Japanese butterfish | 4 |
| Latidae | *Lates calcarifer* | Barramundi/giant seaperch | 4 |
| Nemiptidae | *Nemipterus virgatus* | Golden thread | 3.99 |
| Sciaenidae | *Atrobucca nibe* | Blackmouth croaker | 3.96 |
| Carangidae | *Seriola quinqueradiata* | Japanese amberjack | 3.96 |
| Cynoglossidae | *Cynoglossus bilineatus* | Fourlined tonguesole | 3.89 |
| Lethirinidae | *Lethrinus olivaceus* | Longface emperor | 3.85 |
| Lutjanidae | *Lutjanus vitta* | Snappers/brownstripe | 3.8 |
| Haemulidae | – | Grunt fish | 3.8 |
| Sciaenidae | *Larimichthys crocea* | Large yellow croaker | 3.7 |
| Sparidae | Dentex hypselosomus | Yellowback sea-bream | 3.7 |
| Lethrinidae | *Lethrinus miniatus* | Trumpet emperor | 3.7 |
| Sciaenidae | *Larimichthys polyactis* | Yellow croaker | 3.7 |
| Sparidae | Pagrus major | Red seabream | 3.7 |
| Sparidae | | other seabream | 3.7 |
| Carangidae | *Alepes djedaba* | Shrimp scad | 3.58 |
| Triglidae | *Chelidonichthys ischyrus* | – | 3.5 |
| Macrouridae | *Coelorinchus formosanus* | Formosa grenadier | 3.5 |
| Sciaenidae | *Johnius distinctus* | – | 3.5 |

 

| Family name | Scientific name | Common name | Trophic level |
| --- | --- | --- | --- |
| Sciaenidae | *Miichthys miiuy* | Mi-iuy croaker | 3.5 |
| Mobulidae | Mobula alfredi | Alfred manta | 3.5 |
| Dussumieriidae | *Etrumeus micropus* | Round herring | 3.5 |
| Menidae | *Mene maculata* | Moonfish | 3.45 |
| Carangidae | *Decapterus kurroides* | Redtail scad | 3.4 |
| Sparidae | *Evynnis tumifrons* | Yellow seabream | 3.4 |
| Mullidae | *Parupeneus barberinus* | Dash-and-dot goatfish | 3.4 |
| Latilidae | *Branchiostegus japonicus* | Horsehead tilefish | 3.4 |
| Emmelichthyidae | *Erythrocles schlegelii* | Japanese rubyfish | 3.4 |
| Carangidae | *Trachurus japonicus* | Japanese jack mackerel | 3.4 |
| Ariidae | *Arius maculatus* | Sea catfish/Spotted sea catfish | 3.36 |
| Stromateidae | *Pampus echinogaster* | Silver pomfret | 3.3 |
| Stromateidae | *Pampus argenteus* | Silver pomfret | 3.3 |
| Sparidae | *Evynnis cardinalis* | Threadfin porgy | 3.27 |
| Sparidae | *Acanthopagrus schlegelii* | Blackhead seabream | 3.24 |
| Sillaginidae | *Sillago asiatica* | Asian sillago | 3.2 |
| Spratelloididae | *Spratelloides gracilis* | Silver-stripe round herring | 3.1 |
| Lateolabracidae | *Lateolabrax japonicus* | Japanese seabass | 3.1 |
| Carangidae | *Parastromateus niger* | Black pomfret | 2.9 |
| Plecoglossidae | *Plecoglossus altivelis* | Ayu sweetfish | 2.8 |
| Dorosomatidae | *Nematalosa japonica* | Japanese gizzard shad | 2.4 |
| Scaridae | – | Parrotfish | 2 |

species constitutes more than 50% of the total catch between Scenario 2 and 3, highlighting the disproportionate influence of the species in the overall trophic dynamics of the coastal and offshore fisheries around the waters of Taiwan. In Scenario 3, the CTL analysis will uncover the relative important key indigenous species which play a prominent role in the trophic structure and ecosystem of the water of Taiwan.

# RESULTS

## Overview of offshore and coastal fishery data

The total catch around Taiwan from 1970–2020 was showed in Fig. 2A. The total catch of offshore fishery showed a rapid increase pattern from 1970–1980, with a high peak in 1981 (260 thousand tons). Then stabilizing around 100–150 thousand tons annually until now (Fig. 2C). The total catch of offshore fishery account for over 80% of the proportion for the study period (Fig. 2B). The mainly fishing gear of offshore fishery around Taiwan were the Taiwanese purse seine, trawling, and the gillnet. The total catch of trawling showed the highest and similar pattern with total catch around 1970–1990 (Fig. 2C). The Taiwanese

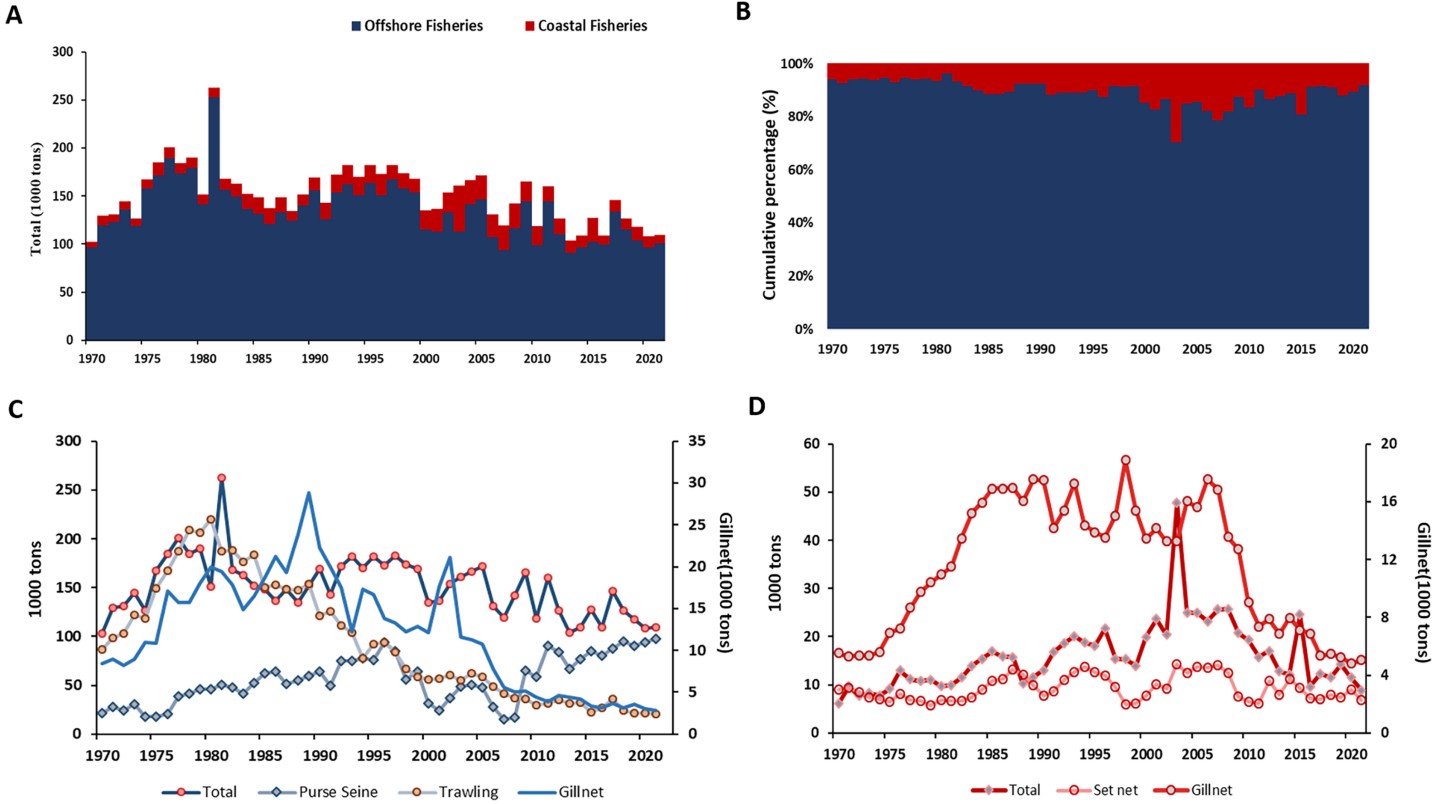

**Figure 2 The time series and cumulative percentage of total catch in coastal and offshore fishery around Taiwan.** (A) The time series (B) cumulative percentage of total catch in coastal and offshore fishery around Taiwan. The total catch with mainly fishing gear of (C) offshore (D) coastal fishery. Data resource: Fisheries Statistical Yearbook from Fishery Agency; www.fa.gov.tw.

purse seine showed total amount lower than trawling. Gillnet showed the lowest value among three mainly fishing gear in Taiwanese offshore.

The pattern of coastal fishery showed continuously increased from 1970 to 1985, reached to the peak in 2003, then decline afterward (Fig. 2D). With the highest (48 thousand tons) and lowest (eight thousand tons) total catches were in 2003 and 1974, respectively (Fig. 2C). The two mainly fishing gear in coastal fisheries around Taiwan were Set net and gillnet. Total catch of gillnet showed continuously increase pattern from 1970–2005, similar with the total fisheries of coastal fisheries (Fig. 2D).

More advanced, the MTL of offshore fishery showed around 3.8–4.0 from 1970 to 2005, then raised to 4.1 afterward (2006–2021). The MTL of coastal fishery showed lower value compared to offshore fishery with increasing pattern from 1970–1980, showed a stable value around 3.8 to 4.0 afterward, except for the years 2015 and 2021 (Fig. 3A). The pattern of FiB index (0–0.34) for offshore fishery increased from 1970–1980, then showed an oscillation between 0.2–0.4 for 1980 to 2021. Compared to offshore fishery, the FiB of coastal fishery showed the higher value. It showed the rapid increase pattern (FiB = 0–1.2) from 1970–2003, then decreased afterward, especially drop to lowest in 2014 (FiB = 0.44) and 2021(FiB = 0.38) (Fig. 3B).

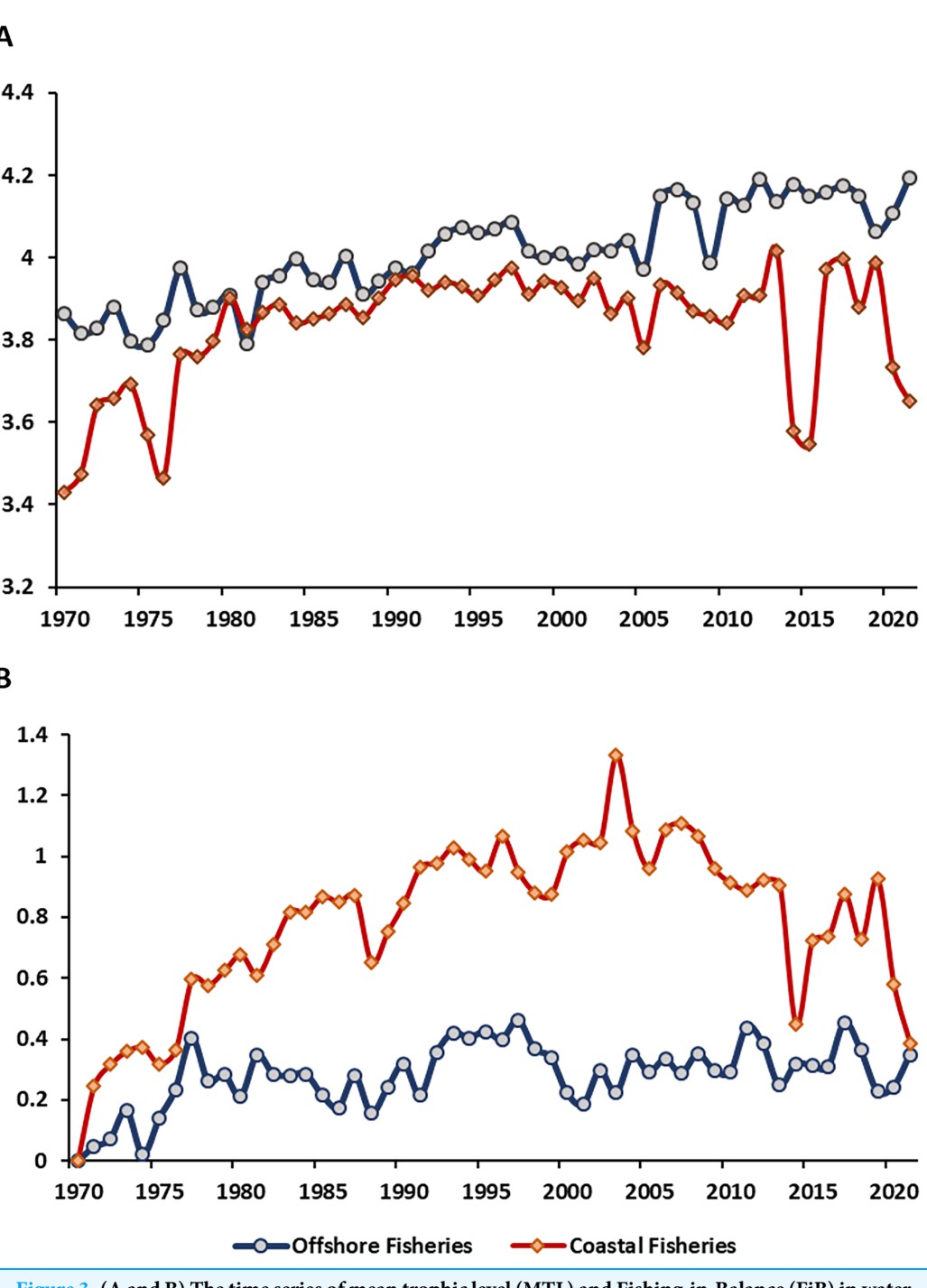

**Figure 3** (A and B) The time series of mean trophic level (MTL) and Fishing-in-Balance (FiB) in water of Taiwan. Data resource: Fisheries Statistical Yearbook from Fishery Agency; www.fa.gov.tw.

## MTL results for each scenario

Scenario 3 has a lower MTL value when compared to Scenario All and Scenario 1 (Fig. 4A). Across all four scenarios from 1971 to 2010, the MTL increase approximately 0.07 to 0.08 per decade on average (Table 4A). Scenario All showed the highest linear $R^2$ value of 0.76

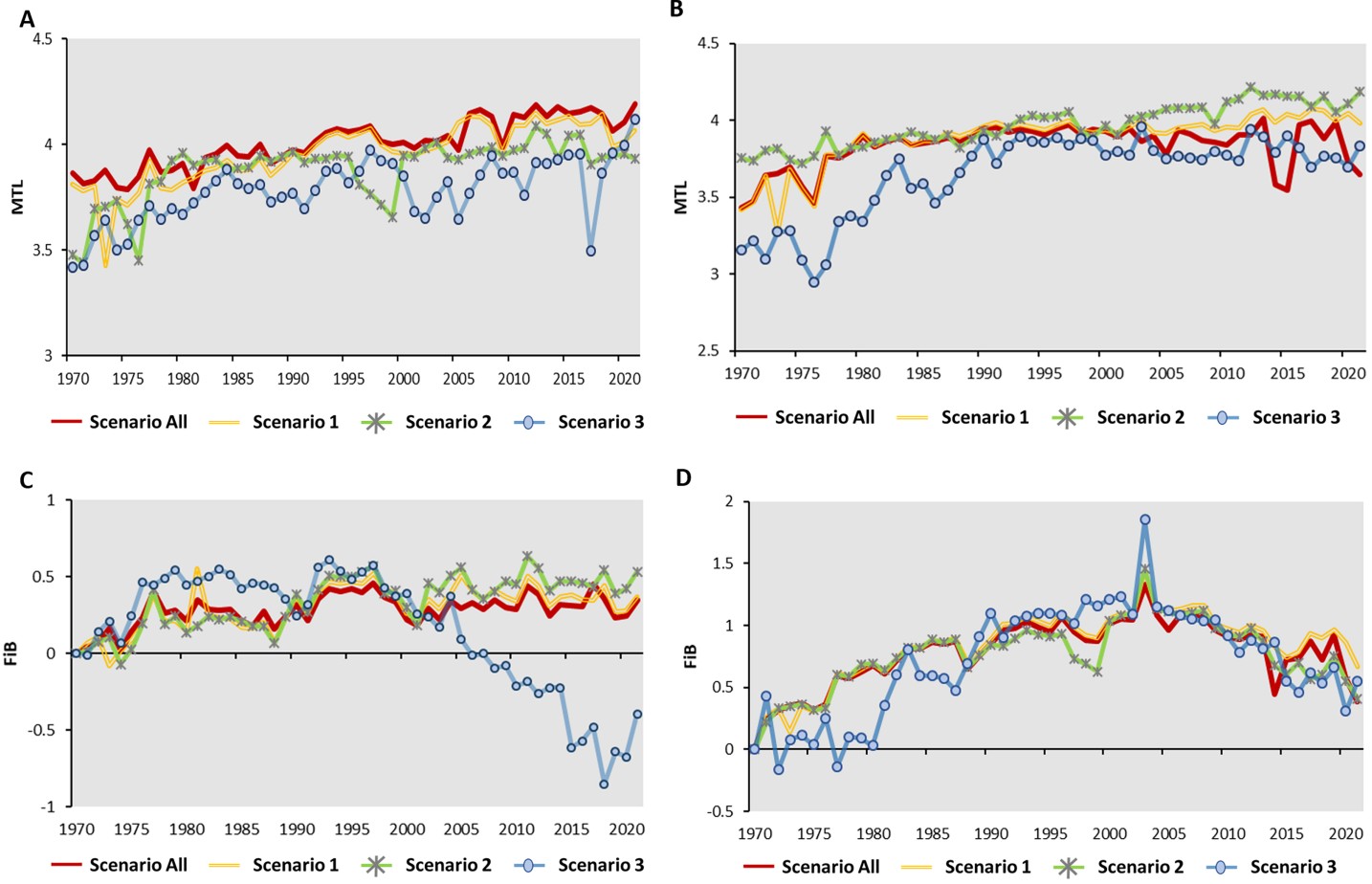

**Figure 4** The time series of MTL for (A) offshore fishery (B) coastal fishery and Fishing-in-Balance (FiB) for (C) offshore fishery (D) coastal fishery. Data resource: Fisheries Statistical Yearbook from Fishery Agency; www.fa.gov.tw.

**Table 4** Measures of patterns in (a) MTL and (b) FiB over time, in fisheries data.

| Scenario | Offshore fisheries | | | Coastal fisheries | | |
|---|---|---|---|---|---|---|
| | Per decade MTL change | Linear $R^2$ | $p$ | Per decade MTL change | Linear $R^2$ | $p$ |
| Scenario all | 0.07 | 0.76 | <0.01 | 0.05 | 0.15 | <0.01 |
| Scenario 1 | 0.08 | 0.69 | | 0.10 | 0.59 | |
| Scenario 2 | 0.07 | 0.40 | | 0.09 | 0.85 | |
| Scenario 3 (Indigenous sp.) | 0.08 | 0.43 | | 0.15 | 0.57 | |
| Scenario | Offshore fisheries | | | Coastal fisheries | | |
| | Per decade FiB change | 1970–2021 Linear $R^2$ | $p$ | Per decade FiB change | 1970–2021 Linear $R^2$ | $p$ |
| Scenario all | 0.04 | 0.25 | <0.01 | 0.07 | 0.24 | <0.01 |
| Scenario 1 | 0.06 | 0.36 | | 0.11 | 0.41 | |
| Scenario 2 | 0.09 | 0.60 | | 0.06 | 0.19 | |
| Scenario 3 (Indigenous sp.) | −0.20 | 0.08 | <0.05 | 0.14 | 0.25 | |

**Note:**
$R^2$ values of linear trend lines, and the significance of the trend, $P$-value.

($p < 0.01$), with a decline in the linear $R^2$ values as species-moved by each scenario. For coastal fisheries, the MTL value ranged from 3.5 to 4.0, except for Scenario 3 (indigenous species) (Fig. 4B). Scenario All showed the gentle pattern with the linear $R^2 = 0.15$ ($p < 0.01$). Following the removed-process, Scenario 1 and 2 showed the increase value around 0.1 per decade on average of MTL with linear $R^2 = 0.59$ and 0.85, respectively ($p < 0.01$) (Table 4A). The Scenario 3 (indigenous species) indicated the most per decade change (0.15) among the four scenarios.

## FiB results for each scenario

The standard year of FiB for the four scenarios was 1970. The FiB for offshore fishery in Scenario All, 1, and 2 showed the similar pattern from 1971–2021. Scenario 3 (indigenous species) showed the higher value than other scenarios during 1975–1990, then showed the dramatically decreased pattern from 1990–2021. The values would vary between 0 and 0.5 for the study period in Scenario All, 1, and 2 with per decade variations around 0.04–0.09 (linear $R^2 = 0.25$). The FiB index for Scenario 3 (indigenous species) increased to 0.5 from 1970 to 1975, then remained stable between 1980 and 1995, except for 1990. The value of FiB declined from 0.5 to −1 in the period since 1995 with per decade variation change around −0.2 (Fig. 4C; Table 4B). The pattern of FiB for coastal fishery showed the increased pattern started from 1970 with a high peak in 2003 among the four scenarios then indicated the decline pattern afterward. The FiB for Scenario All, 1, and 2 raised to 1.5 at 2003. Compared to other scenarios, the Scenario 3 (indigenous species) showed higher FiB index (2.0) in the same period (Fig. 4D).

## CTL results for offshore and coastal fishery

To capture the crucial species that influenced the fisheries dynamics, the CTL of offshore and coastal species from Scenario 2 and 3 (indigenous species) were calculated. For Scenario 2, the *Scomber australasicus* (34.1%), *Trichiurus lepturus* (11.5%), and Auxis spp. (6.4%) take significant proportions of the catches for offshore and coastal fishery (Table 5A). Focusing on the annual catches percentage of influential species of offshore fisheries, the *Scomber australasicus* shows an increased in catches from 1980, eventually accounting for almost 80% of the offshore fishery catches. *Trichiurus lepturus* showed a high catch percentage from 1970 to 1990, then started to decrease after that. *Scomberomorus spp.* had notable catch proportions around 1970–1990. *Auxis spp.* and *Polydactylus sextarius* showed the similar catch pattern throughout the time period for offshore fishery (Fig. 5A). For coastal fishery, *Trichiurus lepturus* constitutes a significant portion (around 20–40%) of the total catch compared to the other species. *Scomber australasicus* catches increased from 1980 to 2001, then decreased until the present (Fig. 5B).

For Scenario 3 (indigenous species), the CTL of each importance species did not exceed 10% (Table 5B). The highest CTL species was the *Pennahia argentata* (9.9%), *Priacanthus macracanthus* (8.6%), *Enchelycore schismatorhynchus* (7.4%), and *Saurida elongate* (7.1%) take around 35% CTL for offshore and coastal fishery. For offshore fishery, every important species took around 10–20% catches through the study period (Fig. 6A). For

**Table 5 Fish species that has high contribute trophic level (CTL) to mean trophic level (MTL) and FiB (a) scenario 2 and (b) scenario 3 for offshore and coastal fishery.**

| Species | TL | The average CTL from 1970–2021 |
|---|---|---|
| *Scomber australasicus* | 4.23 | 34.10% |
| *Trichiurus lepturus* | 3.99 | 11.50% |
| *Auxis spp.* | 4.24 | 6.35% |
| *Sardinella sindensis* | 2.90 | 4.30% |
| *Scomberomorus spp.* | 4.35 | 2.24% |
| *Polydactylus sextarius* | 4.06 | 2.23% |
| Total | | 60.72% |
| **Species** | **TL** | **The average CTL from 1970–2021** |
| *Pennahia argentata* | 4.06 | 9.92% |
| *Priacanthus macracanthus* | 4.11 | 8.62% |
| *Enchelycore schismatorhynchus* | 4.38 | 7.42% |
| *Saurida elongata* | 4.5 | 7.06% |
| *Pagrus major* | 3.7 | 5.27% |
| Sparidae | 3.7 | 8.97% |
| *Evynnis tumifrons* | 3.4 | 5.80% |
| *Nemipterus virgatus* | 3.99 | 2.47% |
| Total | | 55.53% |

coastal fishery, *Pennahia argentata* indicated over 20% catch percentage. *Priacanthus macracanthus* and *Enchelycore schismatorhynchus* each accounted around 10% catches through the time (Fig. 6B).

# DISCUSSION

This study collected Fisheries Statistical Yearbook from 1970 to 2021, and found that the offshore fisheries of Taiwan began investing heavily in fishing during the 1980s (Fig. 2A). Consistent with the research that mentioned the number of fishing vessels reaching the highest peak in the 1980s (*Kuo & Booth, 2011*). However, production and total catches of offshore have decreased from 1980 afterward, the decrease of fisheries resource and the emergence of the sustainability concepts have led to the formulation of fisheries-related policies and regulations (Fig. 2C; *Fisheries Agency, 2013*; *Liu et al., 2009*; *Liao, Huang & Lu, 2019*). The coastal fisheries showed the increase pattern from 1970 to 1990, with the consistent pattern to the total catch of gillnet. The regulation of gillnet started to set up by each local county from the beginning of 2000, this might be one of the reasons to show the decreased pattern of total catch for coastal fishery (*Fisheries Agency, 2022*). Another reason is the regulation implemented in 2000 that prohibits large trawling vessels (over 50 tons) from entering to coastal area (*Fisheries Agency, 2021*).

The MTL indicator of offshore and coastal fisheries showed continuously increase pattern (per decade change between 0.05–0.01) through the time (Table 5A; Figs. 4A, 4B). Several studies also pointed out the continuously increase MTL pattern in Central Amazon

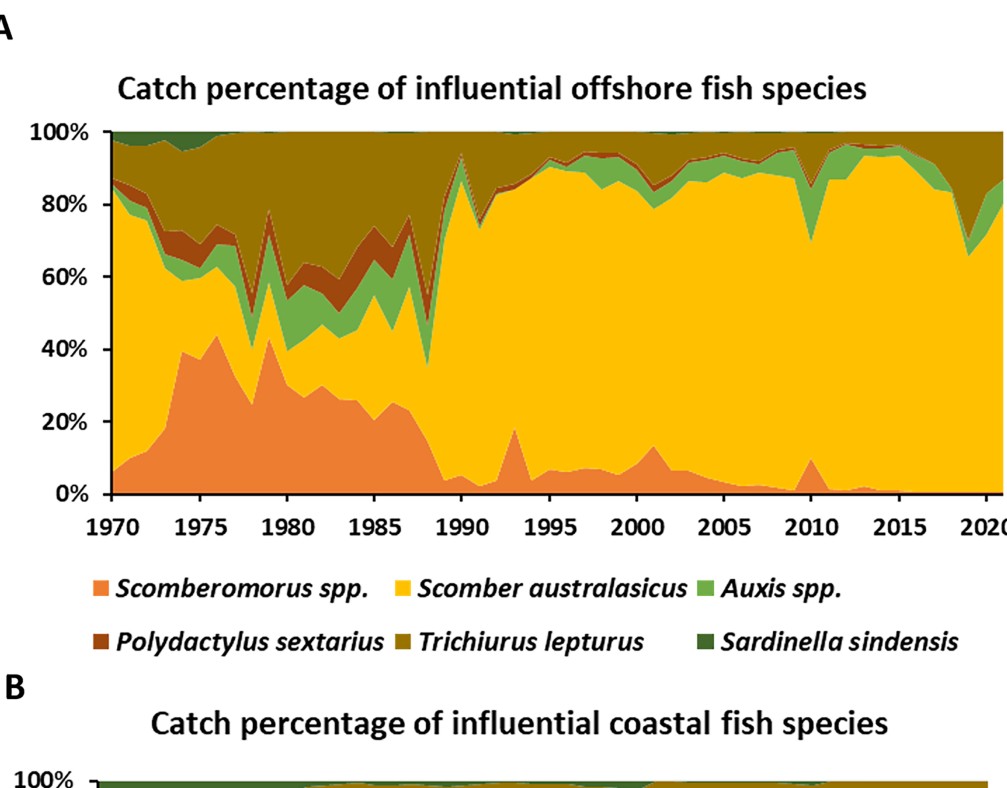

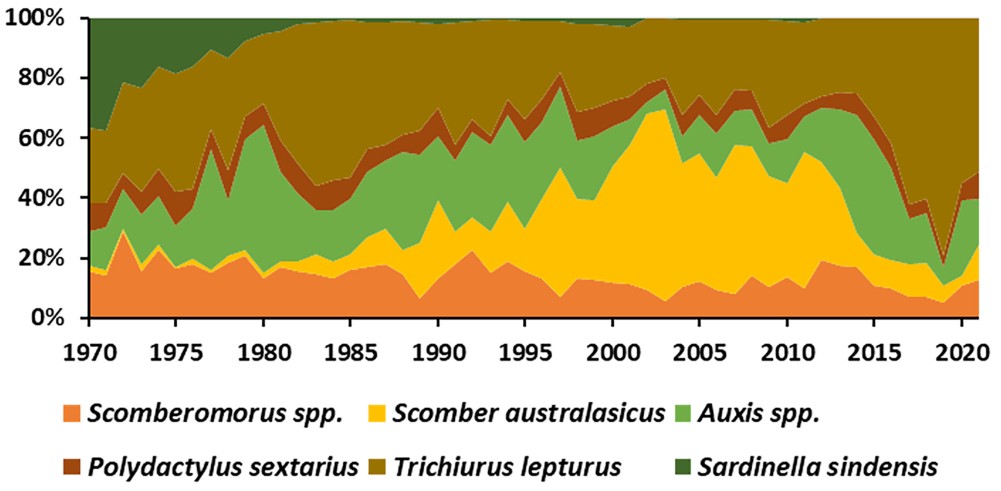

**Figure 5** The catch percentage variations of influential species with high CTL of (A) offshore and (B) coastal fisheries in Scenario 2.

and Changshan Islands around China (*Cao, Sun & Yang, 2023*; *Matos et al., 2024*). Previous studies pointed out that fishery resources in the Persian Gulf and Oman Sea exhibited an over-exploited status, as indicated by a decrease in the Mean Trophic Level (MTL) and an increase in the Fisheries Biodiversity (FiB) index from 1997 to 2016 (*Hashemi & Geasemza, 2019*; *Razzaghi, Mashjoor & Kamrani, 2017*). Both MTL and FiB indicators in China's marine area also showed a slight increase from the 1950s to the 1980s, leading to changes in the body sizes of small yellow croaker, largehead hairtail, and Bombay duck (*Bo & Qisheng, 2004*; *Du et al., 2015*). Most studies suggest that advanced

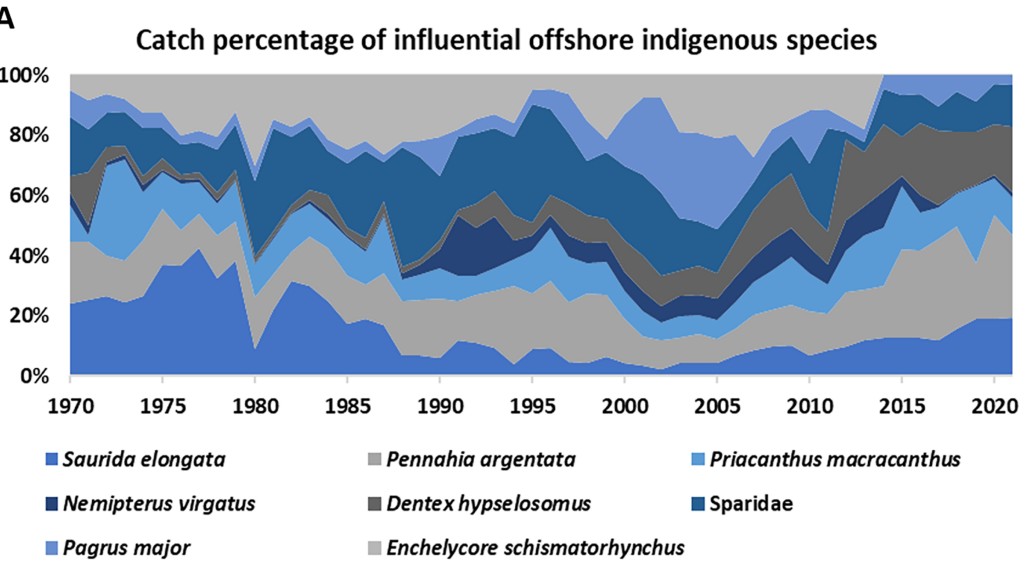

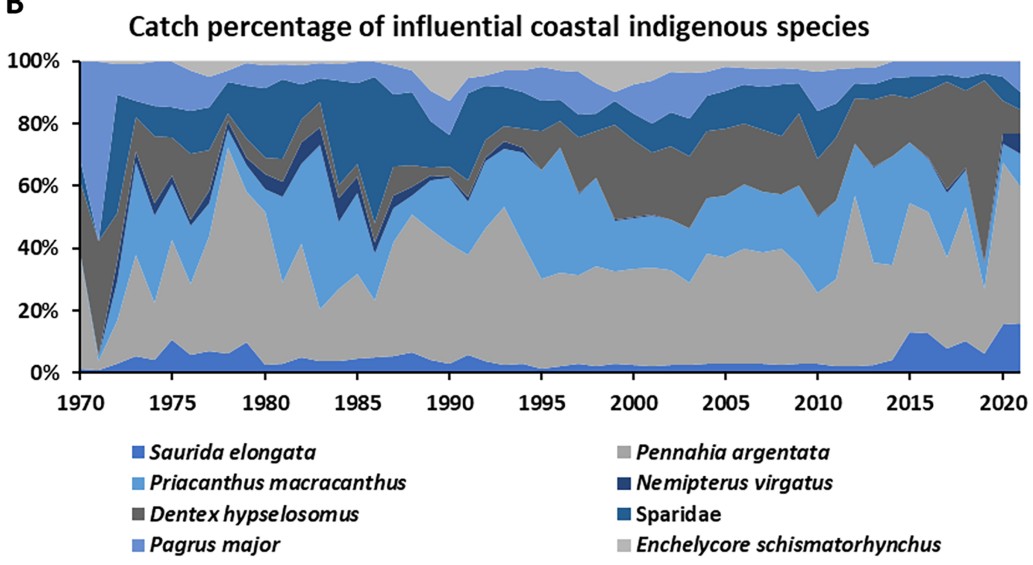

**Figure 6 (A and B) The catch percentage variations of influential species of offshore and coastal for Scenario 3, the indigenous species with high CTL of offshore and coastal fisheries around water of Taiwan.** Data resource: Fisheries Statistical Yearbook from Fishery Agency; www.fa.gov.tw.

investigation into the ecological impacts (*e.g.*, fish body length, landings, *etc.*) on MTL and FiB indicators will provide a more comprehensive understanding of local fishery resource dynamics. There are also lack of the study discussing the offshore and coastal fishery dynamic of waters of Taiwan. Further advance discussion and evaluation are needed in the following steps. There are also lack of the study discussing the offshore and coastal fishery dynamic of waters of Taiwan. Further advance discussion and evaluation are needed in the following steps.

For Scenario All and 1 for offshore and coastal fishery in this study may contribute to an increase in MTL with the highly valuable species (*e.g.*, tuna or billfish) and highly trophic levels (*Benetti, Partridge & Stieglitz, 2016*). In Taiwan, tuna is mainly caught by longline in offshore fisheries and is highly valuable for the Taiwanese fishery economy (*Liu et al., 2024*). Compared to offshore fisheries, coastal fisheries primarily target other oceanic and seasonal migratory species (*e.g.*, chub mackerel, dolphinfish, or white croaker) using set nets (*Fishery Research Institute, 2003*). The focus on oceanic and seasonal migratory species caused Scenario 2 to show a slightly higher MTL than Scenarios All and 1 (Fig. 4B). To Summarize all above-mentioned information, the pattern of MTL showed continuously increase pattern in three scenarios for offshore and coastal fishery around Taiwan. This increasing MTL trend is concerning as expansion of fisheries (*Maureaud et al., 2017*), it may indicate as a potential unsustainable situation and excessive exploitation of local fishery resources dynamics, which could lead to long term ecological impacts (*Christensen, 2000*). Specifically, targeting higher trophic levels can disrupt the food web balance, leading to cascading effects on marine biodiversity and ecosystem stability (*Maureaud et al., 2017*).

## Offshore fisheries

In Scenario All, 1, and 2, the FiB index of offshore fishery showed values between 0–0.5 throughout the study period, with a per-decade change ranging from 0.04 to 0.09 (Fig. 4C; Table 5B). The positive and increasing FiB value may indicate the expansion of fishing grounds to target highly valuable migratory species (*Pauly, Christensen & Walters, 2000*). The CTL analysis and MTL indicator indicated the mackerel species *Scomber australasicus* targeted by Taiwanese Purse Seine was a major factor to cause higher value in Scenario 2. According to *Sun et al. (2006)*, advancements in Taiwanese Purse Seine technique after 1985 led to a significant increase the total catch of Purse Seine (Fig. 2C). Therefore, the main reason influenced the variation of FiB index in Scenario 2 was the development of Taiwanese Purse Seine fishing technique for fishing mackerel.

In Scenario 3, the FiB index showed the per decade change for −0.2 caused by the decreasing pattern from 1990 to 2020 (Fig. 4C; Table 5B). Based on the positive FiB values from 1970 to 1980, the expansion of fishing grounds to capture commercially valuable indigenous species is also evident in Scenario 3 (*Pauly, Christensen & Walters, 2000*). Since 1980, the Taiwanese Fisheries Agency has been formulating fisheries-related policies and regulations focused on sustainability in response to the continuous decrease in total catch (*Fisheries Agency, 2013*; Fig. 2). Consistent to the regulation of trawling fisheries that formulate between 1990 and 2017 response to a continuous decline in the total catch of trawling fisheries, so did the decreasing FiB index and fishing resource.

However, the FiB index did not stabilize around 0, especially showing a declining trend after 2003. As shown by the increasing MTL in Fig. 4 and Table 5A indicated that fishing activities continued to target higher trophic level indigenous species, and suggests that offshore fisheries may be experiencing a fish-down mechanism. The *Saurida elongata* dominated from 1970–1980, then *Nemipterus virgatus* (TL = 3.99) started to replace to become the mainly caught species for Taiwanese offshore fisheries until 2000 (Fig. 7). *Pagrus major* (TL = 3.7) and *Dentex hypselosomus* (TL = 3.7) also became the dominant

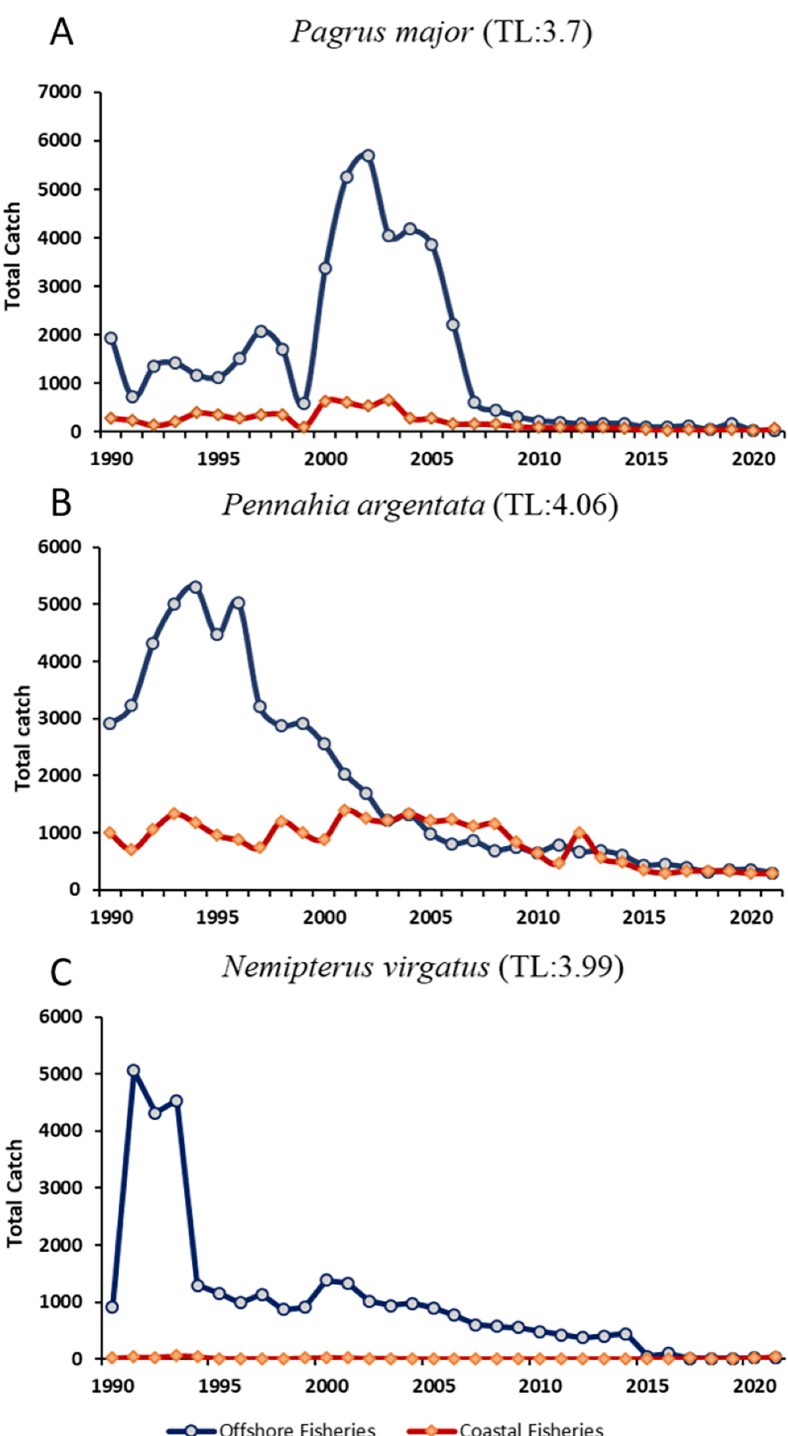

**Figure 7** (A–C) The time series of three key indigenous species that extracted from Scenario 3, *Pagus major, Pennahia argentata, Nemipterus virgatus.* Data resource: Fisheries Statistical Yearbook from Fishery Agency; www.fa.gov.tw.               

catches after 2003. These shifts in species composition further emphasize the need for adaptive management strategies to mitigate the impacts of overfishing and ensure the sustainability of fisheries (*Mcleod et al., 2019*). The ecological impact of such shift includes

potential depletion of key species, altering predator prey relationships, and reducing genetic diversity, which can affect the resilience of marine ecosystems to environmental changes.

## Coastal fisheries

The pattern of FiB showed the consistent pattern across all four scenarios with the similar pattern of the total catch for gillnet fishing (Figs. 2D, 4D). This point out fishing gear was the mainly issue to influence the pattern of FiB index for coastal of Taiwan compared to significantly species. Gillnets are the predominant fishing gear used along the coast of Taiwan, and designed to capture species across entire trophic levels with minimal emphasis on targeting specific species, excluding the grey mullet gillnet (*Lan et al., 2017*; *Tseng & Kao, 2022*). This characteristic of fishing gear makes coastal fishery resource dynamics dependent not only on specific significant species. The coastal fishery started to develop and increased the fishery production from 1970–1990 which having the same trend with the offshore fisheries (*Kuo & Booth, 2011*). However, the continuous decline in the FiB index implicated the fisheries resources might not operate at a sustainable level (*Pauly, 2010*).

The resource dynamics and the regulation formulation of coastal fisheries worthy to attention. *Liao, Huang & Lu (2019)* pointed out the complex composition of fishing gear in Taiwanese coastal fishery lead to poor management. In response to rising awareness of sustainability, Taiwanese Fisheries Agency started to issue laws and regulations to manage the fishery resource. Trawling, Taiwanese Purse Seine, and gillnet with multiple layers are prohibited from entering the area within 12 nm offshore started in 2000 (*Fisheries Agency, 2016*). The comprehensive management of the fishery resource will be a crucial for the Taiwanese government to maintain ecological balance and ensure long-term sustainability.

## Compare and explain the result between Sea Around Us and our results

This study highlights the importance of continuous monitoring and assessment of fishery resources to inform effective conservation and management interventions. To achieve sustainable fisheries, it is essential to adopt adaptive management strategies that incorporate scientific research and ecosystem-based approaches. Sea Around Us (https://www.seaaroundus.org/) also developed the globally MTL and FiB index to monitor the biodiversity loss, including the waters around Taiwan. From 1974–2019, the MTL values reported by Sea Around Us ranged between 35 and 3.7, and the FiB value showed a stable pattern from 1970 to 2010. In Scenario 3 (indigenous species) of our study, the MTL values showed the similar pattern to those reported by Sea Around Us (Fig. 8). This similarity supports the idea that removing tuna and other migratory species is necessary for a clearer assessment of the coastal fishery resource. However, the FiB values in the four scenarios before 2010 are higher than those reported from the Sea Around Us project. This discrepancy may be due to different potential conditions (spatial expansion of fisheries; *Kleisner, Mansour & Pauly, 2015*) considered in the fishery dynamic evaluation of the surrounding waters of Taiwan. This comparison underscore the need for enhanced data

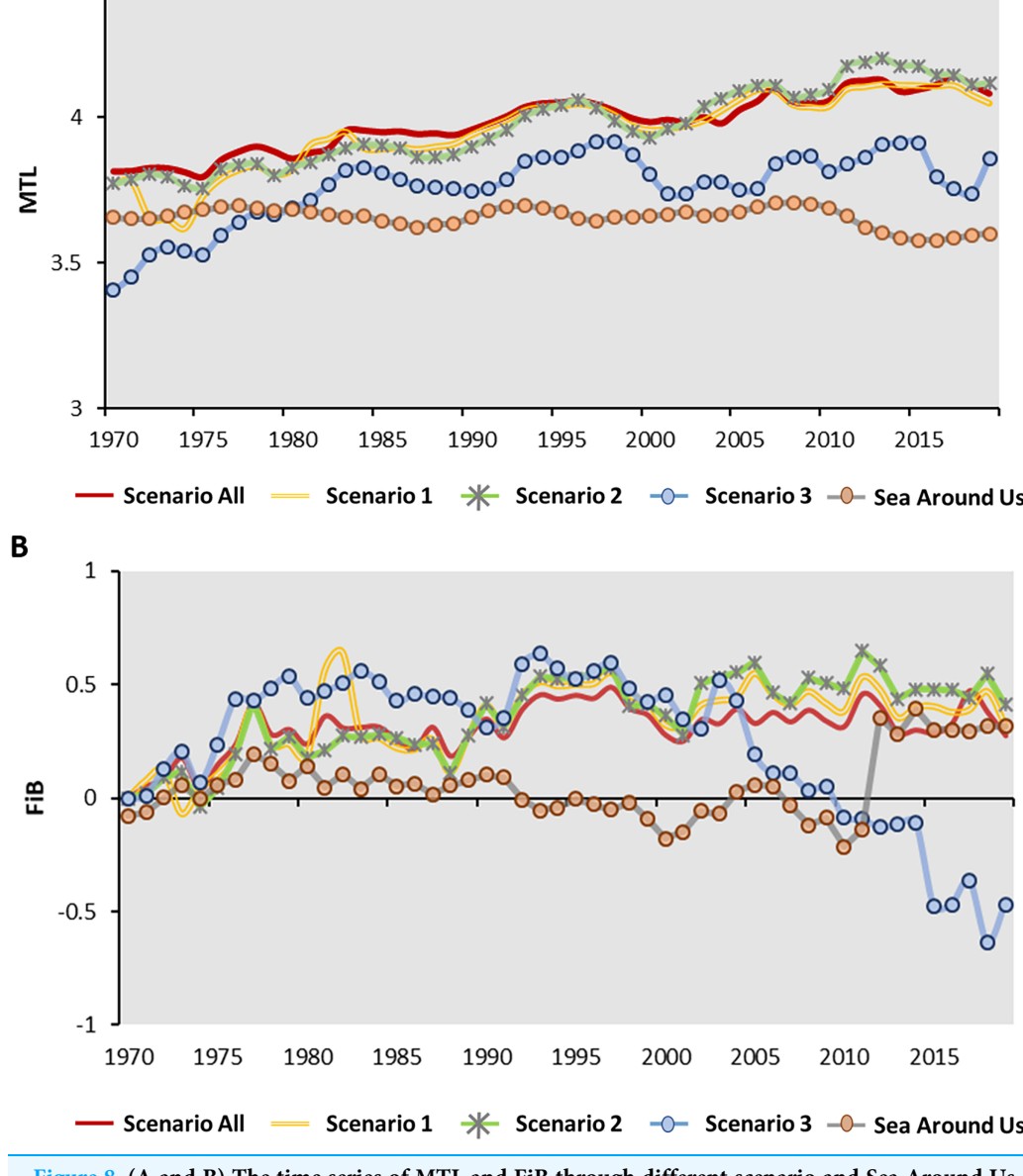

**Figure 8** (A and B) The time series of MTL and FiB through different scenario and Sea Around Us webpage from 1970 to 2019. Data resource: Fisheries Statistical Yearbook from Fishery Agency; www.fa. gov.tw and Sea Around Us: https://www.seaaroundus.org.

collection and international collaboration to improve the accuracy and effectiveness of fishery management strategies.

## CONCLUSION

This study established various scenarios for evaluating MTL and FiB to gain a nuanced understanding of how fish species and fishing practices affect fishery dynamics of coastal and offshore fisheries in Taiwan. The variations in the catch of crucial species of offshore fisheries, particularly in Scenario 2 and Scenario 3, were found to directly impact the MTL

and FiB index. This highlights the importance of these species in the overall fishery dynamics and underscores their critical role in maintaining the ecological balance of Taiwan's waters. This finding revealed the continuous negatives of FiB value in offshore fishery, suggesting potential issues related to trawling fishery and the behavior of fishing-down mechanism. This indicates a possible over-exploitation of fish stocks, which may lead to long-term unsustainable practices if not addressed. Although the Taiwanese government is committed to promoting the fishing-gear related policies, and has tried to investigate the fishing dynamic to achieve the goal of SDG14. There are still several issues (*e.g.*, the more detail of multiple fishing gear policy) to address before achieving sustainable development of fisheries resources.

To enhance fisheries management in Taiwan, it is essential to focus on the impact of different fishing gears and also the number of power craft belonged to each fishing gear on resource dynamics. Policymakers should consider implementing stricter regulations on fishing gear use, such as setting quotas or restrictions, to mitigate over-exploitation, alongside promoting sustainable fishing practices through incentives and engaging stakeholders in policy development which ensure practical and socio-economically sensitive strategies. Further studies on fishing vessels and fisheries yield management will be crucial for sustaining Taiwan's fishery resources, The findings of this study underscore the needs for continuous monitoring and adaptive management practices, which are vital for long-term sustainability of Taiwan's offshore and coastal fisheries. This can be achieved by investing in research to understand the ecological impact of different fishing practices. In conclusion, this study provides a comprehensive understanding of the fishery dynamics in Taiwan and offers valuable insights for developing strategies to ensure the sustainability of fishery resources. By addressing the identified issues and implementing adaptive management strategies, Taiwan can move towards achieving its goals for sustainable fisheries and marine conservation.

### Funding
This work was supported by the Taiwan Fishery Agency (112 fishery management-2.38-policy-02, and 113 fishery management 2.37-coastal-02) and the National Science and Technology Council (112-2611-M-019 -023 and NTSC112-2811-M-019 -016). The funders had no role in study design, data collection and analysis, decision to publish, or preparation of the manuscript.

### Grant Disclosures
The following grant information was disclosed by the authors:
Taiwan Fishery Agency.
National Science and Technology Council: 112-2611-M-019 -023 and NTSC112-2811-M-019 -016.

## Competing Interests

The authors declare that they have no competing interests.

## Author Contributions

- Yan-Lun Wu conceived and designed the experiments, performed the experiments, authored or reviewed drafts of the article, and approved the final draft.
- Irene Chia Ling Lim conceived and designed the experiments, analyzed the data, prepared figures and/or tables, and approved the final draft.
- LiXliang Li performed the experiments, authored or reviewed drafts of the article, and approved the final draft.
- Lu-Chi Chen analyzed the data, authored or reviewed drafts of the article, and approved the final draft.
- Po-Yuan Hsiao conceived and designed the experiments, analyzed the data, prepared figures and/or tables, and approved the final draft.
- Wei-Yu Lee performed the experiments, analyzed the data, authored or reviewed drafts of the article, and approved the final draft.
- Kuo-Wei Lan conceived and designed the experiments, performed the experiments, analyzed the data, prepared figures and/or tables, authored or reviewed drafts of the article, and approved the final draft.

## Data Availability

The raw measurements are available in the Supplemental Files.

## Supplemental Information

Supplemental information for this article can be found online at http://dx.doi.org/10.7717/peerj.18434#supplemental-information.

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
