# Peer review of "The evolution of resource management in Taiwanese fisheries: coastal and offshore perspectives"

_PeerJ, doi:10.7717/peerj.18434_

## Round 0.1 · original submission · Minor Revisions

Dear Dr. Wu

The reviewers have commented on your manuscript. You can find the attached reports. Based on the comments and suggestions of the expert reviewers, a minor revision is needed for your article.

I would like to request you check and correct the manuscript step by step based on the reports.

Sincerely yours

·

Basic reporting

-Language is clear used throughout this manuscript.
-Background is enough to overview the topic.
-Manuscript structure is in good flow.
-Raw data were shared.

Experimental design

-The authors state that anchovy species are not included in the calculations and they are right in this regard, but they should specify the catch amounts of these species so that the reader can have a good idea of what has been left out.
-The authors should provide detailed information on the Fisheries Statistical Yearbook. is this a digital data source? has it improved over the years?
-The authors mention the fishing fleet in many places, but details such as the number of boats according to fishing techniques, fleet size are missing. If the number of boats by years is available, it should be mentioned.
-The study area should be shown on a map. If possible, a distinction between onshore and offshore areas should be shown to integrate the spatial understanding of the findings.
-Which program was used to produce the graphs?

Validity of the findings

-The data are robust and statistically sound.

-The comparison with "Sea Around Us" in the discussion section is quite logical, but the reader does not have enough information about "Sea Around Us" in the introduction.

Reviewer 2 ·

Basic reporting

While the English used in the manuscript is generally proficient, there are a few grammatical errors that need attention.

I believe that additional literature references would strengthen the work.
The structure of the article is sound; however, some of the figures have resolution issues that should be addressed.
The raw data is well-presented, and the hypotheses have been effectively explained with appropriate visualizations. Nevertheless, certain analyses and graphs could benefit from further refinement and development."

Experimental design

The study is original and aligns well with the aims and scope of the journal.
The research question is well-defined and addresses a regional gap; however, there are deficiencies in its presentation. Additionally, the methods have not been described with sufficient detail and clarity.

Validity of the findings

The conclusions are well articulated, but there are some shortcomings.

Annotated reviews are not available for download in order to protect the identity of reviewers who chose to remain anonymous.

Reviewer 3 ·

Basic reporting

The manuscript entitled “Exploring the evolution of resource development in Taiwanese
offshore and coastal fsheries” includes a fifty year data set, which is already sufficient for a scientific paper. I see that MS is well written with a clear and understandable English. Structure of the MS is also well organised. However, abstract and discussion sections need to be improved. Abstract should be more informative and the main theme of the study should be more evident.And for the disccussion section, it must be widen in order to bring more ideas to discuss. I recommend authors to include more references in this section so it will hep them to compare not only coastal and offshore fisheries, but also different geographical locations. Authors must be correct with the writing rules for scientific names of the species (italic font and the author of each species).
Consequently, manuscript is well organised, contains new and significant information to justify publication and content of the study is stunning. Manuscript is not only scientifically valuable but also informative for public. So, my decision is accept with a minor revision.

Experimental design

Well organised and clear

Validity of the findings

Findings are presented well with the supporst of figures and tables.

---

## Round 0.2 · accepted · Accept

Dear Dr. Wu

I would like to thank you and your co-authors for making the corrections and changes requested by the reviewers. I read and checked your valuable article carefully and am happy to inform you that your article has been accepted for publication in PeerJ.